# Vaginal Microbiota in Short Cervix Pregnancy: Secondary Analysis of Pessary vs. Progesterone Trial

**DOI:** 10.3390/diseases13100338

**Published:** 2025-10-14

**Authors:** Antonio G. Amorim Filho, Roberta C. R. Martins, Lucas A. M. Franco, Juliana V. C. Marinelli, Stela V. Peres, Rossana P. V. Francisco, Mário H. B. Carvalho

**Affiliations:** 1Disciplina de Obstetricia, Departamento de Obstetricia e Ginecologia, Faculdade de Medicina, Universidade de Sao Paulo, Sao Paulo 05403-000, Brazil; jucodato@gmail.com (J.V.C.M.); svperes80@gmail.com (S.V.P.); rossana.francisco@hc.fm.usp.br (R.P.V.F.); mario.burlacchini@hc.fm.usp.br (M.H.B.C.); 2Instituto de Medicina Tropical, Faculdade de Medicina, Universidade de Sao Paulo, Sao Paulo 05403-000, Brazil; betacristina@gmail.com (R.C.R.M.); lucasamfranco@hotmail.com (L.A.M.F.)

**Keywords:** premature labor, cervix uteri, bacterial vaginosis, metagenome, microbiota, vaginal discharge, progesterone, pessary

## Abstract

Background/Objectives: Preterm birth (PTB) is a leading cause of neonatal mortality, particularly in women with a short cervix. Vaginal dysbiosis has been associated with increased PTB risk. Progesterone (PR) and Arabin pessary (PE) are commonly used for PTB prevention, but their impact on vaginal microbiome composition is unclear. This study aimed to compare the effects of these interventions on the vaginal microbiome in women at risk of PTB. Methods: In a secondary analysis of a randomized trial at Hospital das Clínicas, Universidade de São Paulo, 203 women with singleton pregnancies and cervical length ≤ 25 mm at the second trimester were assigned to daily vaginal PR (200 mg) or PE. Vaginal swabs from 44 participants (*n* = 22 per group) were collected at baseline and 4 weeks post-treatment and analyzed via 16S rRNA gene sequencing. Results: From 88 samples analyzed, 80 showed a low-diversity, *Lactobacillus*-dominated microbiota, 42 classified into *Lactobacillus iners*-dominated community state type (CST-III), and 38 presented other *Lactobacillus* species dominance (termed CST-I/II/V). The remaining eight samples presented non-*Lactobacillus* dominance (CST-IV). Comparing the two groups, no significant changes in CST were observed between sampling timepoints (PE group, *p* = 0.368; PR group, *p* = 0.223). Similarly, Shannon alpha diversity did not change (PE group, *p* = 0.62; PR group, *p* = 0.30), and Bray–Curtis dissimilarity also did not change after treatment (*p* = 0.96, before; *p* = 0.87, after treatment). Conclusions: Arabin pessary and vaginal progesterone maintain vaginal microbiome stability in women at high PTB risk, supporting the microbiological safety of both interventions.

## 1. Introduction

Preterm birth (PTB), defined as birth before 37 weeks of gestation, is a primary public health concern owing to its significant contribution to infant morbidity and mortality. In Brazil, the incidence of PTB is approximately 11.5%, accounting for at least three-quarters of infant mortality and often leading to long-lasting sequelae, such as cerebral palsy and learning disability, in survivors [1]. Despite the complex interplay of risk factors and causes, spontaneous PTB (sPTB), which occurs without any medical indications, is associated with bacterial colonization of the amniotic cavity via an ascending vaginal infection. Bacterial vaginosis (BV) is one such risk factor [2]. The precise mechanism remains unclear, but it is believed that bacterial vaginosis (BV) promotes the release of inflammatory mediators, such as cytokines and prostaglandins, which contribute to cervical ripening and the onset of uterine contractions. In addition, vaginal dysbiosis, such as BV, may compromise the protective barrier provided by the normal microbiota, making the environment more susceptible to infection and preterm birth (PTB) [2]. Therefore, investigating vaginal dysbiosis is crucial for understanding the pathophysiology of spontaneous preterm birth (sPTB). For example, Carvalho et al. [3] analyzed a cohort of 611 pregnancies at the same institution as in the present study and found a significantly higher incidence of sPTB in women with BV than in healthy women (9.7% vs. 3.2%, *p* = 0.008). Another important risk factor is the presence of a short cervix (i.e., ≤25 mm), typically identified via transvaginal ultrasound (TVUS) screening during the second trimester of gestation [4]. Daily vaginal progesterone (PR) tablets are a well-established therapy for preventing sPTB in pregnant women with short cervical length. Although not fully understood, the protective effects of progesterone are thought to result from both the inhibition of uterine contractions and its anti-inflammatory properties [5]. On the other hand, the Arabin pessary (PE), a flexible silicone ring placed within the vagina, has a purely mechanical effect, supporting the cervix and maintaining its integrity. It has been proposed as an alternative treatment, and it is being investigated in clinical trials; however, results remain inconsistent [6]. Conde-Agudelo et al. [7], in a recent meta-analysis involving 12 studies (4687 women), concluded that current evidence does not definitely support the use of a cervical PE to prevent sPTB in pregnant women with a short cervix. Thus, further research is warranted, particularly in pregnancies at a higher risk of sPTB (such as in women with a short cervix [i.e., ≤25 mm], a history of sPTB, or women with a very short cervix [i.e., <10 mm]). Despite this uncertainty, the use of a PE for sPTB prevention is routine in many settings may be a less expensive alternative for the Brazilian public health system. It is generally well tolerated but can sometimes be associated with worsened or concerning vaginal discharge [6,8]. Cervical cerclage, a surgical procedure that closes the cervix with stitches, is reserved for patients at risk of cervical insufficiency [9].

Microbiome studies employing 16S rRNA gene sequencing techniques have shown that the vaginal ecosystem in asymptomatic, reproductive-age women typically falls into five compositions, based on the abundance of microbial taxa, also termed basic community state types (CSTs). Four CSTs are characterized by a low-diversity, *Lactobacillus*-dominated microbiota, distinguished by the dominant *Lactobacillus* species: *Lactobacillus crispatus* (CST-I), *Lactobacillus gasseri* (CST-II), *Lactobacillus iners* (CST-III), and *Lactobacillus jenseni* (CST-V). In contrast, the fifth CST (CST-IV) is more diverse and lacks *Lactobacillus* dominance, often featuring BV-associated anaerobes [10]. The distribution of these CSTs varies considerably by ethnicity and physiological condition [11]. For instance, during uncomplicated pregnancies, the vaginal microbiome typically becomes more stable, less diverse, and dominated by *Lactobacillus* species [12,13,14,15], which are believed to confer protection against infectious microorganisms [16]. By contrast, studies [17,18,19,20,21,22] have shown that sPTB is associated with vaginal microbiome dysbiosis, characterized by loss of stability, increased diversity, and reduced *Lactobacillus* content. Of note, CST-IV profiles (i.e., low in *Lactobacillus* content) are more prevalent in ethnic groups at higher risk for sPTB, including Hispanic and African American women [15]. Current evidence supports an association between low-*Lactobacillus* CSTs and sPTB risk, as well as the protective effect of *L. crispatus* CST-I [23,24]. *L. crispatus* maintains a stable and healthy vaginal environment through various mechanisms, including the production of D-lactic acid and L-lactic acid (D-La and L-La, respectively) isomers, which reduce vaginal pH and inhibit microbial proliferation. D-La, in particular, may reduce inflammation in the vaginal mucosa, contributing to its stability. By contrast, *L. iners*-dominated communities (CST-III) primarily produce L-La, which lacks anti-inflammatory properties, thereby making the environment prone to dysbiosis and potential adverse effects [25].

In this study, we aimed to investigate—using microbiome profiling of vaginal secretions from pregnant women at high risk of sPTB who were treated with a PE or PR—whether the treatment itself could drive dysbiosis associated with sPTB, thereby posing further threats to these pregnancies. This association has been evaluated in only a few studies [19,26]; therefore, we believe this information will be valuable in helping clinicians choose an appropriate treatment for sPTB. A preprint of the present study has previously been published [27].

## 2. Materials and Methods

### 2.1. Study Design and Participants

This study was part of an ongoing single center randomized trial comparing the efficacy of PE and PR treatment for sPTB prevention in women with singleton pregnancies and short cervical lengths who were diagnosed via TVUS during the second trimester (NCT02511574, *clinicaltrials.gov*, Bethesda, MD, USA). The trial was conducted at the prenatal care clinic of Hospital das Clinicas, Universidade de São Paulo (São Paulo, Brazil). Sample size was calculated based on the rate of sPTB before 34 weeks and the local prevalence of a short cervix during the second trimester [4]. By using a logistic regression model, we estimated a 20% reduction in the sPTB rate with PE or PR with a type 1 error (α) of 5% and a statistical power (1 − β) of 80% [4]. Two-hundred and three participants were accordingly randomized to receive 200 mg of vaginal PR daily or a 65/25/35 mm Arabin PE (Dr. Arabin GmbH and Co., Witten, Germany). The inclusion criteria were singleton pregnancies at 20–24 weeks of gestation; a cervical length of ≤25 mm, assessed with TVUS; intact membranes; no history of cervical insufficiency; no painful uterine contractions; and no major fetal abnormalities. Only women with singleton pregnancies were included because no consensus has been reached on the effect of PE or PR on twin pregnancies. PR treatment was discontinued, or the PE was removed at ≤37 weeks in cases of sPTB.

As an exploratory sub-analysis of the RCT aimed at investigating whether the treatment was associated with vaginal microbiome dysbiosis, a cohort of 44 individuals (22 from each treatment group) was selected via convenience sampling for microbiome profiling. The microbiome composition was determined using 16S rRNA gene amplicon sequencing [10] at two timepoints: before treatment at randomization (*T*_0_) and after 4 weeks of treatment (*T*_1_) at the first subsequent patient visit (*T*_1_), as per protocol. Previous studies in which sampling was conducted at similar timepoints have shown that vaginal CSTs remain relatively stable between the second and third trimesters in uncomplicated pregnancies [14,19]. Notably, in the study by Kindinger et al. [19], involving a cohort of pregnant women at high risk of sPTB treated with vaginal progesterone, a significant difference in CST composition was observed at 28 weeks between those who delivered before and after 34 weeks. Therefore, the four-week sampling interval appeared adequate for detecting treatment-induced changes in CSTs. To ensure our methods produced results consistent with the expected microbiome composition in normal pregnancies, vaginal samples from four pregnant women with similar baseline characteristics but a normal cervical length (i.e., >25 mm) were initially profiled (Figure A1).

For baseline data analysis, categorical variables were compared using the chi-square or Fisher’s exact test when frequencies of <5% were observed. Continuous variables were compared using the Mann–Whitney *U* test or Student’s *t*-test, depending on whether the data were distributed non-normally or normally, respectively. Statistical analyses were performed using the SPSS software package (v24.0; IBM Corp., Armonk, NY, USA). A significance level of 95% was considered for all analyses. The baseline characteristics of the participants were similar for the two study groups (Table 1). The study was conducted in accordance with the Declaration of Helsinki and approved by the Ethics Committee for Analysis of Research Projects (CAPPesq) of the Hospital das Clinicas, Universidade de São Paulo (São Paulo, Brazil; approval number: CAAE 20611813.0.0000.0068; approval date: 27 November 2013).

### 2.2. Sample Collection, DNA Extraction, Library Preparation, and DNA Sequencing

Vaginal specimens were collected during routine prenatal care visits via speculum examination. Samples were obtained using a standard plastic brush, dispersed in 2 mL of sterile 0.9% saline, and immediately stored at −80 °C for subsequent analysis. A second sample was collected simultaneously in an appropriate transfer medium for Gram staining and Nugent scoring [28]. Bacterial DNA was extracted from the thawed samples in a clean, sterile environment by using a PowerSoil kit (MoBio Laboratories, Carlsbad, CA, USA), based on the manufacturer’s recommendations.

The V4 region of the 16S rRNA gene was amplified via polymerase chain reaction (PCR), using primers F515 (5′-CACGGTCGKCGGCGCCATT-3′) and R806 (5′-GGACTACHVGGGTWTCTAAT-3′) [29]. Sequencing adapters and barcodes were added to the primers (full primer sequences are summarized in Table A1). PCR amplification was conducted with the Platinum PCR SuperMix High Fidelity kit (Thermo Fisher, Waltham, MA, USA) under the following cycling conditions: 94 °C for 3 min, followed by 30 cycles of denaturation at 94 °C for 30 s, annealing at 58 °C for 30 s, and extension at 68 °C for 1 min. Amplicons were purified and quantified, as previously described [30]. Templates were prepared using the Ion Chef System (Thermo Fisher Scientific, Waltham, MA, USA), and DNA sequencing was performed using the Ion Personal Genome Machine (Thermo Fisher Scientific) with 318 semiconductor chips, based on the manufacturer’s recommendations.

### 2.3. Microbiome Profiling of the Vaginal Samples

Microbiome analysis was conducted using Quantitative Insights into Microbial Ecology (QIIME 2; version 2020.2; Caporaso Lab at Northern Arizona University, Flagstaff, AZ, USA). Raw sequences were demultiplexed, denoised, and truncated at 240 bp, using a minimum Phred score of 33 to generate amplicon sequence variants (ASVs) by using the DADA2 algorithm [31]. The average number of ASVs per sample was 232.965 (range: 55.588–544.234). ASVs were taxonomically classified into operational taxonomic units (OTUs) with a minimum of 97% similarity, and a phylogenetic tree was constructed using Greengenes 13_8 (https://greengenes2.ucsd.edu/#, UCSD, San Diego, CA, USA) as the reference. The data were subsequently summarized in a feature table containing the relative abundance of OTUs at each taxonomic rank.

Microbial communities were clustered into CSTs by constructing a dendrogram with Microbiome Analyst (https://www.microbiomeanalyst.ca), using the complete linkage algorithm and Bray–Curtis dissimilarity as distance metrics (Figure A2). Given the limited species-level resolution of the V4 16S rRNA sequencing strategy, it was not possible to differentiate CST-I (*L. crispatus*), CST-II (*L. gasseri*), and CST-V (*L. jensenii*) with confidence. These were therefore grouped together as CST-I/II/V (i.e., non-*L. iners*-dominant communities). Importantly, CST-III (*L. iners* dominance) and CST-IV (non-*Lactobacillus*-dominant communities) remained clearly distinguishable, allowing for meaningful clinical interpretation.

### 2.4. Statistical Analyses of Microbiomes

The CST profiles of each treatment group were compared between *T*_0_ and *T*_1_ using the McNemar–Bowker test. Rarefied samples were analyzed using QIIME2 to determine alpha (species richness and Shannon index) and beta diversity indices (i.e., Bray–Curtis dissimilarity). Alpha diversity was compared between the PE and PR groups at both timepoints, using the Kruskal–Wallis test (performed with the R package v4.2.0), following the syntax provided by Chen et al. [32]. Beta diversity comparisons were conducted using permutational multivariate analysis of variance–pairwise analysis, with visualizations generated through Principal Coordinates Analysis plot by using EMPeror (Microbiome Analysis and Visualization Lab, University of California, San Diego, San Diego, CA, USA) [33]. All statistical analyses were conducted at a significance level of 95%. Differential taxonomic abundance was determined using analysis of the composition of microbiomes (ANCOM) within QIIME2.

## 3. Results

The mean age of patients was 30.1 ± 7.2 years in the PE group and 28.1 ± 6.0 years in the PR group. Most participants were White (68.2% and 72.7%, respectively), and 15.9% reported a history of sPTB. In the current pregnancy, sPTB occurred in 9.1% of patients in the PE group and 30.0% in the PR group. While this difference may seem clinically relevant, it did not reach statistical significance. For reference, the original RCT reported sPTB rates of 32.4% and 29.7% in the PE and PR groups, respectively (*p* = 0.68). Overall, baseline characteristics were consistent with those of the RCT, with no significant differences between groups. Nonetheless, the small sample size of the present study should be acknowledged as a limitation (Table 1).

As shown in Figure 1 and consistent with expected trends during pregnancy, most analyzed samples exhibited low diversity and were *Lactobacillus*-dominated (*n* = 80). Of these, 42 samples were dominated by *L. iners* (i.e., CST-III), whereas 38 samples were dominated by other *Lactobacillus* species (i.e., CST-I/II/V). The remaining eight samples had higher diversity, no *Lactobacillus* dominance, and increased abundance of anaerobes, including BV-associated *Gardnerella* (i.e., CST-IV), which indicated a state of dysbiosis. The observed CST distribution was consistent with that of previous findings in a larger, similar Brazilian population in which *L. iners* dominance was also common [34].

Most samples in both groups had normal Nugent scores (Table 1). An altered score (i.e., >3), while not frequently observed, was associated with highly diverse communities, as expected (Figure 1).

With regard to community stability, some CST changes were observed between the two sampling timepoints, with two samples (AF10 and AF71) shifting toward the potentially dysbiotic CST-IV. Notably, sample AF10 was collected from a patient in the PE group who delivered at 25 weeks. In contrast, sample AF71 was obtained from the PR group, and the patient delivered at term. Yet, no significant difference in the CST profile was observed between *T*_0_ and *T*_1_ in the PE group (*p* = 0.368) or PR group (*p* = 0.223) (Figure 2). Thus, treatment with a PE or PR did not significantly alter the CST distribution. *L. iners* dominance similarly did not increase with either treatment.

A comparison of individual sample species diversity, or alpha diversity, was then performed by using two methods: richness (i.e., observed), which accounts solely for the number of observed species in the community, and the Shannon index, which accounts for the richness and the evenness of species distribution. Comparisons between *T*_0_ and *T*_1_ within each group (Figure 3A,B) and between each group at *T*_0_ and *T*_1_ (Figure 3C,D) revealed no significant differences. This finding suggested that both groups were equally diverse and stable during the observation period.

No significant differences in species composition dissimilarity or in beta diversity were observed between the sampling timepoints in the PE or PR group (Figure 4) because no clusters related to the sampling timepoint were observed in the ordination plot. The Bray–Curtis dissimilarity method was chosen because of its quantitative nature, making it more suitable for the analysis of communities dominated by a specific species such as the vaginal microbiome. This method was also preferred for CST clustering in the study population (Figure A2). Additional diversity analyses were performed using other methods. A complete dataset can be found at https://drive.google.com/drive/folders/12uEFU2X0AuTpqToRhMOuoMJnSOYYTvpp?usp=drive (accessed 29 September 2025).

Finally, differential abundance analysis using ANCOM, which accounts for the compositional nature of microbiomes, was performed. It allows for the inference of the absolute abundance from the relative abundance and for the correction of sampling bias. This analysis was conducted to further explore possible differences in taxa abundance between treatment groups and sampling timepoints, including dominant features such as *Lactobacillus* species and *Gardnerella*. Accordingly, ANCOM did not detect any differences in taxa abundances between *T*_0_ and *T*_1_ in either the PE or PR group (Figure A3). Although the small sample size and the known limitations of ANCOM must be considered, these findings suggest that neither treatment was associated with significant changes in the abundance of individual taxa.

## 4. Discussion

In this study, we compared the vaginal microbiome after PE or PR treatment in women at high risk for sPTB. We found that, regardless of the treatment method, the vaginal microbiome remained stable in terms of bacterial diversity and CST composition, similar to patterns observed in normal pregnancies [12,13,14,15]. Vaginal PR is the standard treatment for sPTB prevention in women with a short cervix. It is generally regarded as safe. Moreover, by promoting proliferation of *Lactobacillus* and maintenance of an acidic pH, PR helps in keeping a safe vaginal environment [16]. By contrast, a PE functions as a foreign body and can potentially increase the susceptibility of the vaginal environment to colonization by infectious microorganisms and consequently increase the risk of sPTB. Indeed, PE is frequently associated with abundant and malodorous vaginal discharge. However, in our study, neither treatment appeared to drive vaginal dysbiosis more than the other, indicating that both treatments are equivalent from the perspective of the vaginal microbiota. Considering the mechanisms of action, microbiome stability was expected for PR, but comes as a surprise for PE, as no correlation with the characteristics of vaginal contents was observed.

Our findings are consistent with those of two previous studies addressing this issue. Kindinger et al. [19] observed no PR-associated alterations in the vaginal microbiome in a predominantly Caucasian English population. Vargas et al. [26] studied a mixed-ancestry European cohort and similarly found that patients treated with cerclage (another vaginal foreign body) had higher vaginal microbiome diversity and a reduced *Lactobacillus* content; however, women treated with a PE did not exhibit such microbiome modifications. A possible explanation for this difference could be related to the component materials of the devices. Maybe the silicone used to make PE renders it less prone to pathogen proliferation.

Considerable efforts have recently been expended to identify sPTB-related signatures in the vaginal microbiome. Growing evidence suggests that sPTB may be associated with vaginal ecosystem instability [35]. Microbiome-based sPTB prediction models have been developed using large publicly available datasets. However, the compositional complexity and heterogeneity of the molecular and statistical techniques employed across studies have made drawing definite conclusions difficult [36]. *Lactobacillus* dominance, CST type, and the presence of specific microorganisms such as *Gardnerella* are clinically key factors to consider when evaluating sPTB risk.

An argument is that the microbial composition associated with a short cervix may already reflect a dysbiotic state, predisposing patients to sPTB, even before treatment begins. For instance, Witkin et al. [34] analyzed the data of a larger Brazilian population (*n* = 340), with most (90%) presenting with a normal-sized cervix. They found the CST distribution was similar to that demonstrated in our present study. An interesting finding is that the authors identified CST-III as a potential risk factor for cervical shortening, possibly because of biochemical modifications related to L-La secretion by *L. iners*, resulting in a proinflammatory state. Kindinger et al. [19] similarly suggested that CST-III may represent an intermediate dysbiotic state. Therefore, CST-III dominance in our population may not only reflect a high risk of sPTB but also be a causative factor in cervical shortening. This observation warrants further investigation. Screening pregnant women for CST-III may be clinically relevant and contribute to sPTB prevention in the future. Additionally, adjunctive strategies aimed at modulating the vaginal microbiota toward a more eubiotic (i.e., healthy) composition may improve sPTB prevention when used with PE or PR treatment.

A key strength of our study lies in its focus on vaginal microbiome stability in a Brazilian population at risk of sPTB. This patient group has not been extensively studied in this context. Moreover, to the best of our knowledge, the present study is the first in which PE and PR treatments were compared. In previous studies, each of these treatments was independently compared against a no-treatment control (e.g., cervical length > 25 mm).

Nevertheless, this study has several limitations. The relatively small sample size (*n* = 22 per group), although comparable to those reported by Kindinger et al. [19] (*n* = 25, PR group) and Vargas et al. [26] (*n* = 26, PE group), limits the statistical power to detect microbiome alterations. For instance, in a *post hoc* analysis, the power to detect CST shifts towards dysbiosis was only 25%, assuming a fixed sample of 22, the lowest observed shift rate (9%), and a one-tailed hypothesis (calculated with the software G*Power 3.1.9.7.). Consequently, establishing definitive conclusions about the association between treatment-induced vaginal dysbiosis and sPTB remains challenging. Another limitation of our study relates to the sequencing strategy employed, which intrinsically restricts taxonomic resolution at the species level. This limitation precluded accurate discrimination among CST-I (*L. crispatus*), CST-II (*L. gasseri*), and CST-V (*L. jensenii*). Consequently, these subtypes were aggregated and analyzed as CST-I/II/V (i.e., non-*L. iners*-dominated profiles). Although this approach was methodologically necessary, it may have underestimated the actual heterogeneity within *Lactobacillus*-dominated CSTs and potentially obscured species-specific associations with clinical outcomes. For instance, previous evidence suggests that *L. crispatus* may confer stronger protection compared with other *Lactobacillus* species. Further analyses of the remaining 159 samples from the original trial, together with larger prospective cohorts employing higher species-level resolution, will be critical to validate our observations and to determine their potential clinical implications.

## 5. Conclusions

Vaginal PR tablets and Arabin PEs are widely prescribed to prevent sPTB in at-risk women with a short cervix, identified during the second trimester. Our findings indicate that these treatments do not significantly alter the vaginal microbiome; thus, patients and clinicians can be reassured about the safety of these methods with regard to infectious complications. This knowledge contributes to improving the quality of care provided to patients at risk of sPTB.

## Figures and Tables

**Figure 1 diseases-13-00338-f001:**
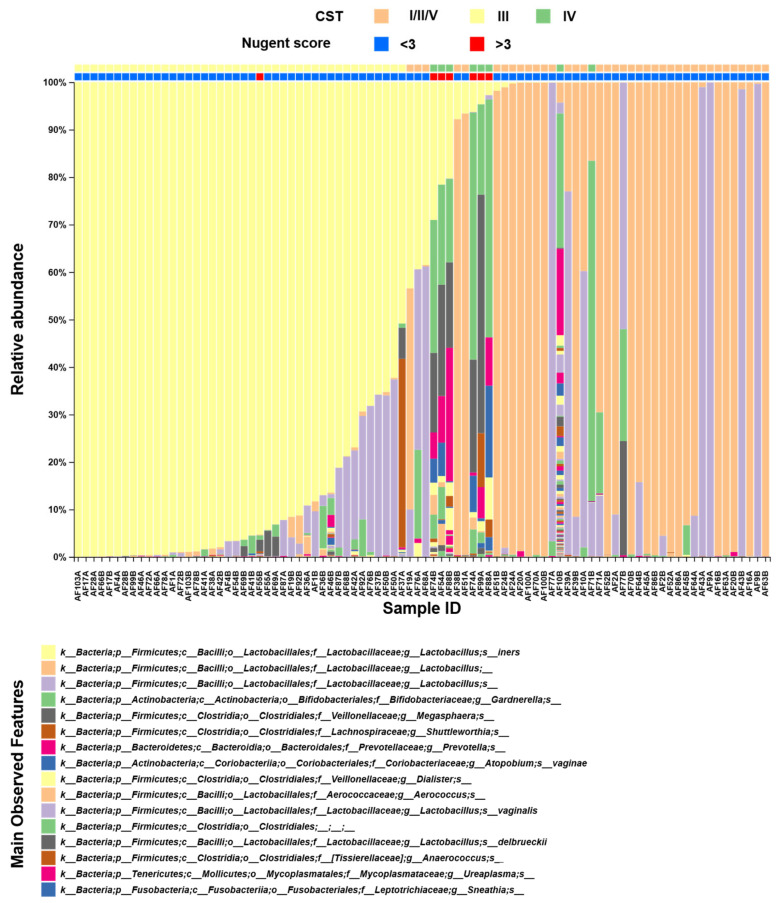
Composition of the vaginal microbiome. The taxa bar plot shows the main features identified, as shown in the bottom panel. Samples were ordinated based on *L. iners* abundance. The letters A and B in each sample identification (ID) denote timepoint *T*_0_ (i.e., before treatment) and timepoint *T*_1_ (i.e., after treatment), respectively. Community state types and Nugent scores are indicated by the bars at the top. CST: community state type; *L. iners*: *Lactobacillus iners*.

**Figure 2 diseases-13-00338-f002:**
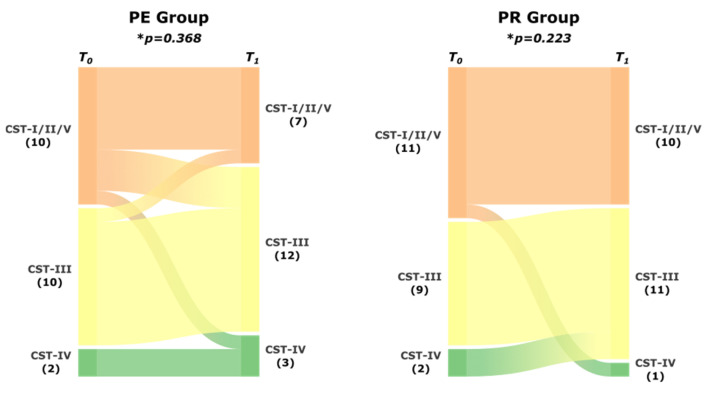
Microbiome stability after treatment for a short cervix. The alluvial diagram shows the comparison of the distribution of the CST before treatment (*T*_0_) and after treatment (*T*_1_) in the PE and PR groups. * Based on the McNemar–Bowker test. PE: pessary; PR: progesterone; CST: community state type.

**Figure 3 diseases-13-00338-f003:**
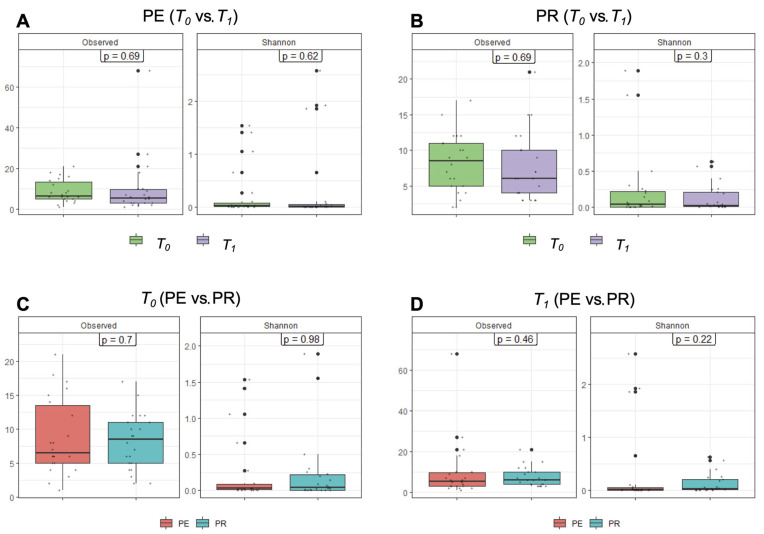
Vaginal microbiome alpha diversity at the genus level. The upper two panels show the comparison of alpha diversity between *T*_0_ and *T*_1_ in the PE group (**A**) and PR group (**B**). The lower two panels show the comparison between the PE and PR groups at *T*_0_ (**C**) and *T*_1_ (**D**). In the boxes, the *p*-values are based on the Kruskal–Wallis test. Observed: richness; Shannon: Shannon’s diversity index; PE: pessary; PR: progesterone; *T*_0_: before treatment; *T*_1_: after treatment.

**Figure 4 diseases-13-00338-f004:**
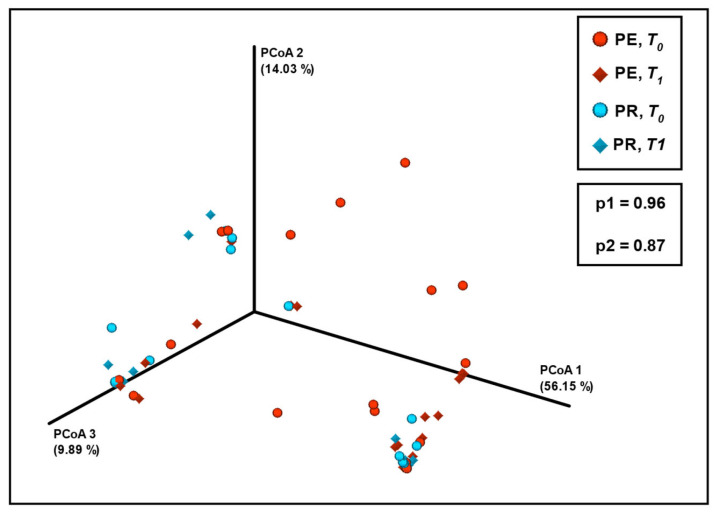
The PCoA-ordinated plot of vaginal microbiome beta diversity, using the Bray–Curtis dissimilarity method at the genus level. The red dots indicate the PE group at *T*_0_ (i.e., before treatment); the red diamonds indicate the PE group at *T*_1_ (i.e., after treatment). The blue dots indicate the PR group at *T*_0_, and the blue diamonds indicate the PR group at *T*_1_. Bottom box presents the *p*-values, based on the PERMANOVA–pairwise test comparing *T*_0_ and *T*_1_. p1: PE group; p2: PR group. PCoA: principal coordinate analysis; PE: pessary; PR: progesterone; PERMANOVA: permutational multivariate analysis of variance.

**Table 1 diseases-13-00338-t001:** Baseline characteristics of the study population.

Characteristic	PE (*n* = 22)	PR (*n* = 22)	*p*
**Demographics**			
Age, y, mean ± SD (range)	30.1 ± 7.2 (15–42)	28.1 ± 6.0 (17–37)	0.315 ^1^
Ethnicity			
White, *n*	15 (68.2%)	16 (72.7%)	0.741 ^2^
Black/mixed, *n*	7 (31.8%)	6 (27.3%)	
BMI (kg/m^2^), mean ± SD (range)	26.5 ± 4.3 (18.6–36.1)	27.7 ± 5.1 (20.5–37.7)	0.453 ^3^
**Obstetric history**			
Nulliparous, *n*	13 (59.1%)	8 (36.4%)	0.131 ^2^
Miscarriage, *n*	7 (31.8%)	8 (36.4%)	0.750 ^2^
Preterm birth, *n*	4 (18.2%)	3 (13.6%)	1.000 ^4^
**Sample collection**			
Mean GA at T_0_ ± SD (range), wk	22.3 ± 1.1 (20.6–23.9)	22.9 ± 0.8 (21.6–24.9)	0.107 ^3^
Mean cervical length at *T*_0_ ± SD (range), mm	16.0 ± 6.0 (5.0–24.0)	17.0 ± 5.0 (7.0–23.0)	0.494 ^3^
Nugent score > 3, ***n***	4 (18.2%)	3 (13.6%)	1.000 ^4^
**Pregnancy outcome**			
Mean GA at birth ± SD (range), wk	37.4 ± 4.0 (25.4–40.3)	37.4 ± 3.4 (25.9–40.7)	0.677 ^3^
Spontaneous preterm birth, *n*	2 (9.1%)	6 (30.0%)	0.123 ^4^

PE: pessary group; PR: progesterone group; BMI: body mass index; GA: gestational age; SD: standard deviation. ^1^ Based on Student’s *t*-test. ^2^ Based on the chi-square test. ^3^ Based on the Mann–Whitney *U* test. ^4^ Based on Fisher’s exact test.

## Data Availability

The original data presented in this study are openly available in Google Drive at https://drive.google.com/drive/folders/12uEFU2X0AuTpqToRhMOuoMJnSOYYTvpp?usp=drive (accessed 29 September 2025).

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
