# Peer review of "Vaginal Microbiota in Short Cervix Pregnancy: Secondary Analysis of Pessary vs. Progesterone Trial"

_diseases, 2025, doi:10.3390/diseases13100338_

Round 1

Reviewer 1 Report

Comments and Suggestions for Authors

General Comment:
This is a well-structured and clinically relevant study examining the potential effects of Arabin pessary (PE) and vaginal progesterone (PR) on vaginal microbiota in pregnant women at risk for spontaneous preterm birth (sPTB) due to short cervical length. The use of 16S rRNA sequencing and longitudinal sampling provides valuable insights. However, there are several areas where clarification and additional data would improve the manuscript.

1. Study Design and Reporting of Full Cohort Results
While the authors describe a randomized controlled trial enrolling 200 participants, the current manuscript presents microbiome results from only 44 patients (22 per group). It is unclear whether this analysis represents a pre-specified secondary aim or an exploratory substudy.
Recommendation: Please clarify whether this is a secondary analysis of the larger RCT. Additionally, the authors should report delivery outcomes (e.g., sPTB rates, gestational age at delivery) for the entire randomized cohort (N=200), even if briefly, to contextualize the microbiome findings within the broader clinical trial.

2. Sampling Bias and Representativeness
The microbiome substudy analyzed a subset (44 of 200) participants selected by convenience sampling. Without comparison to the remaining participants, there is potential for sampling bias.
Recommendation: Provide a comparison of baseline demographic and clinical characteristics (e.g., maternal age, parity, cervical length, ethnicity, pregnancy outcomes) between the microbiome subset (N=44) and the rest of the cohort (N=156) to evaluate representativeness and mitigate concerns about selection bias.

3. Sample Size Justification for Microbiome Arm
The rationale for selecting 22 patients per group is not discussed. Given that this is a small sample, it is important to explain the choice.
Recommendation: Please provide justification for the microbiome substudy sample size. Was the study powered to detect meaningful differences in CST transitions or diversity indices? A post hoc power analysis or reporting of effect sizes would strengthen confidence in the negative findings. 

Also, the sample size of control group (N=4) could be further discussed to ensure the sample size is sufficient to serve as a representative "normal" state. 

4. Clarity on CST Definitions and Grouping
The authors classify microbial communities into three CSTs (I/II/V, III, and IV) but do not explain this grouping early in the results section. Since prior literature suggests that CST-I (L. crispatus) may be more protective than CST-V (L. jensenii), this grouping may obscure clinically relevant distinctions.
Recommendation: Clearly state in the methods or early results section that due to limitations of the 16S V4 region, species-level resolution between CST-I, II, and V was not possible, and that these were grouped accordingly. Discuss any implications this may have for interpretation.

Minor suggestions:

Several typos and grammar issues were observed:

“Yet,, no significant difference...”  with extra comma

"treatment-induced changes in CSTs.. " with extra period

“contribute for sPTB prevention” → should be “contribute to”

Author Response

The authors sincerely thank the reviewer for the time dedicated to this careful evaluation and for the valuable recommendations provided.

Comment 1: Study Design and Reporting of Full Cohort Results
While the authors describe a randomized controlled trial enrolling 200 participants, the current manuscript presents microbiome results from only 44 patients (22 per group). It is unclear whether this analysis represents a pre-specified secondary aim or an exploratory substudy.Recommendation: Please clarify whether this is a secondary analysis of the larger RCT. Additionally, the authors should report delivery outcomes (e.g., sPTB rates, gestational age at delivery) for the entire randomized cohort (N=200), even if briefly, to contextualize the microbiome findings within the broader clinical trial.

Response to comment 1: We appreciate the reviewer’s insightful comment. We agree that our study constitutes an exploratory sub-analysis of an RCT, in which the sample size was determined by convenience (line 119, highlighted in red). The main results of the original RCT have been submitted to the American Journal of Obstetrics & Gynecology MFM and are currently under peer review. For greater clarity, we have also incorporated the outcomes of the original RCT into the manuscript (line 210, highlighted in red).

Comment 2: Sampling Bias and Representativeness
The microbiome substudy analyzed a subset (44 of 200) participants selected by convenience sampling. Without comparison to the remaining participants, there is potential for sampling bias.
Recommendation: Provide a comparison of baseline demographic and clinical characteristics (e.g., maternal age, parity, cervical length, ethnicity, pregnancy outcomes) between the microbiome subset (N=44) and the rest of the cohort (N=156) to evaluate representativeness and mitigate concerns about selection bias.

Response to comment 2: Thank you for pointing this out. We acknowledge that convenience sampling may introduce bias. To address this, we have added a statement clarifying that baseline characteristics were comparable to those of the larger RCT (line 211, highlighted in red). As mentioned above, the main RCT results are currently under peer review at the American Journal of Obstetrics & Gynecology MFM and will be publicly available once published. We also wish to emphasize the exploratory nature of our study, which was designed to investigate microbiome shifts rather than establish correlations with sPTB.

Comment 3: Sample Size Justification for Microbiome Arm
The rationale for selecting 22 patients per group is not discussed. Given that this is a small sample, it is important to explain the choice.
Recommendation: Please provide justification for the microbiome substudy sample size. Was the study powered to detect meaningful differences in CST transitions or diversity indices? A post hoc power analysis or reporting of effect sizes would strengthen confidence in the negative findings. 

Also, the sample size of control group (N=4) could be further discussed to ensure the sample size is sufficient to serve as a representative "normal" state. 

Response to comment 3: Thank you for your valuable comment. We acknowledge that our study is underpowered to detect significant microbiome changes and, therefore, should be more appropriately regarded as an exploratory analysis. We have revised the Discussion section to highlight this limitation more explicitly. In addition, we included a post hoc analysis to further address this point and provide additional context for the interpretation of our findings (lines 119 and 343, highlighted in red). Regarding the analysis of 4 samples from patients with normal cervical length, we would like to clarify that, as a secondary analysis of a randomized trial comparing progesterone versus pessary, there was no control group in the original trial’s design. The vaginal microbiome profiling of those patients was performed solely with the purpose of assuring that our methods produced results consistent with the expected in normal pregnancies. The text was updated to better clarify this issue (line 131, highlighted in red).

Comment 4: Clarity on CST Definitions and Grouping
The authors classify microbial communities into three CSTs (I/II/V, III, and IV) but do not explain this grouping early in the results section. Since prior literature suggests that CST-I (L. crispatus) may be more protective than CST-V (L. jensenii), this grouping may obscure clinically relevant distinctions.
Recommendation: Clearly state in the methods or early results section that due to limitations of the 16S V4 region, species-level resolution between CST-I, II, and V was not possible, and that these were grouped accordingly. Discuss any implications this may have for interpretation.

Response to comment 4: We agree that this point required clarification. Accordingly, we have revised the Methods section (line 185, marked in red) to address this issue more explicitly. In addition, we have expanded the Discussion to highlight the clinical implications of this methodological limitation (line 347, highlighted in red).

Comment 5: Minor suggestions:

Several typos and grammar issues were observed:

“Yet,, no significant difference...”  with extra comma

"treatment-induced changes in CSTs.. " with extra period

“contribute for sPTB prevention” → should be “contribute to”

Response to comment 5: Thank you, all errors were corrected accordingly.

Reviewer 2 Report

Comments and Suggestions for Authors

Thank you for inviting me to review this manuscript. My specific concerns regarding the paper are included below:  

- The background section of the abstract is too general. It starts with PTB morbidity/mortality and then immediately jumps to vaginal dysbiosis, but the link is not well contextualized. The rationale for specifically testing Arabin pessary vs progesterone in relation to dysbiosis is not fully established.

- A sharper statement like ‘to compare the effect of Arabin pessary and vaginal progesterone on vaginal microbiome composition’ would be stronger for the objective.

- ‘The chi-square, McNemar‒Bowker, and Kruskal‒Wallis tests and permutational multivariate analysis of variance were used to compare the two sampling timepoints.’ It is not necessary to write this sentence in the abstract.

- Reporting no changes in the abstract is vague and does not provide even basic numbers, proportions, or p-values. Readers cannot gauge effect size or statistical robustness.

- Although the main RCT was powered for PTB outcomes, there is no power calculation for microbiome endpoints. With only 22 per group, the study is severely underpowered to detect subtle microbiome shifts.

- The control group of only 4 women is very weak; it cannot provide meaningful comparative insights or validation of the CST distribution.

Author Response

The authors wish to thank you very much for taking the time to review this manuscript.

Comment 1: The background section of the abstract is too general. It starts with PTB morbidity/mortality and then immediately jumps to vaginal dysbiosis, but the link is not well contextualized. The rationale for specifically testing Arabin pessary vs progesterone in relation to dysbiosis is not fully established.

Response to comment 1: We agree, the abstract was re-written to better describe the background and rationale (highlighted in red).

Comment 2: A sharper statement like ‘to compare the effect of Arabin pessary and vaginal progesterone on vaginal microbiome composition’ would be stronger for the objective.

Response to comment 2: We agree, the statement was added as suggested (line 15, highlighted in red).

Comment 3: ‘The chi-square, McNemar‒Bowker, and Kruskal‒Wallis tests and permutational multivariate analysis of variance were used to compare the two sampling timepoints.’ It is not necessary to write this sentence in the abstract.

Response to comment 3: We agree, those sentences were removed from the text.

Comment 4: Reporting no changes in the abstract is vague and does not provide even basic numbers, proportions, or p-values. Readers cannot gauge effect size or statistical robustness.

Response to comment 4: We acknowledge that the results part is indeed vague. Actual result numbers were provided for clarity (line 24, highlighted in red). The whole abstract was re-written to address the reviewer's comments, while maintaining conciseness (250 words). 

Comment 5:  Although the main RCT was powered for PTB outcomes, there is no power calculation for microbiome endpoints. With only 22 per group, the study is severely underpowered to detect subtle microbiome shifts.

Response to comment 5: Thank you for your valuable comment. We acknowledge that our study is underpowered to detect significant microbiome changes and, therefore, should be more appropriately regarded as an exploratory analysis. We have revised the Discussion section to highlight this limitation more explicitly. In addition, we included a post hoc analysis to further address this point and provide additional context for the interpretation of our findings (line 343, highlighted in red).

Comment 6: The control group of only 4 women is very weak; it cannot provide meaningful comparative insights or validation of the CST distribution.

Response to comment 6: Thank you for pointing this out, we would like to clarify that, as a secondary analysis of a randomized trial comparing progesterone versus pessary, there was no control group in the original trial’s design. The vaginal microbiome profiling of 4 patients with normal cervical lenght was performed solely with the purpose of assuring that our methods produced results consistent with the expected in normal pregnancies. The text was updated to better clarify this issue (line 131, highlighted in red).

Round 2

Reviewer 2 Report

Comments and Suggestions for Authors

Thank you for the revisions